# A Weighted-Least-Squares Meshless Model for Non-Hydrostatic Shallow Water Waves



Nan-Jing Wu [1], Yin-Ming Su [2], Shih-Chun Hsiao [2,3], Shin-Jye Liang [1,*] and Tai-Wen Hsu [4,5,*]

[1] Department of Marine Environmental Informatics, National Taiwan Ocean University, Keelung 20224, Taiwan; njwu@mail.ntou.edu.tw

[2] Department of Hydraulics and Ocean Engineering, National Cheng Kung University, Tainan 701, Taiwan; n86094275@gs.ncku.edu.tw (Y.-M.S.); schsiao@mail.ncku.edu.tw (S.-C.H.)

[3] Tainan Hydraulics Laboratory, National Cheng Kung University, Tainan 709, Taiwan

[4] Center of Excellence for Ocean Engineering, National Taiwan Ocean University, Keelung 20224, Taiwan

[5] Department of Harbor & River Engineering, National Taiwan Ocean University, Keelung 20224, Taiwan

\* Correspondence: sjliang@mail.ntou.edu.tw (S.-J.L.); twhsu@mail.ntou.edu.tw (T.-W.H.)

**Abstract:** In this paper, an explicit time marching procedure for solving the non-hydrostatic shallow water equation (SWE) problems is developed. The procedure includes a process of prediction and several iterations of correction. In these processes, it is essential to accurately calculate the spatial derives of the physical quantities such as the temporal water depth, the average velocities in the horizontal and vertical directions, and the dynamic pressure at the bottom. The weighted-least-squares (WLS) meshless method is employed to calculate these spatial derivatives. Though the non-hydrostatic shallow water equations are two dimensional, on the focus of presenting this new time marching approach, we just use one dimensional benchmark problems to validate and demonstrate the stability and accuracy of the present model. Good agreements are found in the comparing the present numerical results with analytic solutions, experiment data, or other numerical results.

**Keywords:** non-hydrostatic; shallow water equations; meshless method; weighted-least-squares





## 1. Introduction

In many practical problems of physical oceanography, marine hydrodynamics, and ocean and coastal engineering, the hydrostatic shallow-water-equation (SWE) models are quite satisfactory to predict the water flows in rivers, lakes, estuaries, and seas. However, when it is applied to water wave problems, the hydrostatic assumption in the models is questionable and often the source of errors. A typical case is the simulation of solitary wave propagation in a frictionless channel with a constant water depth. The shape of the free surface profile should always be symmetric, no matter how long the solitary wave propagates. Nevertheless, simulating the case with a hydrostatic SWE model does not produce the correct result [1–3]. As concluded in [4], such kind of error is due to the exclusion of the dispersion term in the hydrostatic shallow water equations. The significance of the non-hydrostatic pressure in water wave simulations is also illustrated in [5–10].

In [11], the water pressure was divided into two parts, the non-hydrostatic pressure, or so called the hydrodynamic pressure, and the hydrostatic pressure. Applying the Keller-box scheme [12] to approximate the vertical distribution of the non-hydrostatic pressure, several wave phenomena in shallow water flows were simulated by solving the non-hydrostatic shallow water equations. It was shown in [13] that in dealing with the dispersion effects of water waves, the non-hydrostatic shallow water equations outperform the Boussinesq equations [14]. In the recent couple decades, many non-hydrostatic SWE models were developed for weakly nonlinear water wave problems by using the grid/mesh-based methods [1–3,11,13,15–19] such as the finite difference method (FDM) and the finite element method (FEM).

Apart from the grid/mesh-based numerical methods, the meshless methods are recently employed in shallow water flow simulations [20–30]. The application of the meshless methods in solving the non-hydrostatic shallow water equations is innovative, because all these meshless works just listed are hydrostatic SWE models, either implicit or explicit.

In this paper, a weighted-least-squares meshless method is developed to solve the non-hydrostatic shallow water equations and to simulate several weakly nonlinear water wave phenomena. A truly explicit predictor-corrector procedure is proposed for the time marching. Unlike those published non-hydrostatic models, the hydrodynamic pressure is obtained straightforwardly rather than from the solution of a time independent Poisson equation which forms a huge global matrix system. Though the non-hydrostatic shallow water equations are two dimensional, due to the focus of this study is on the explicit time marching procedure of the model, only one dimensional cases are considered and illustrated. Four benchmark cases with available analytical solutions, experimental data as well as other numerical results are used to validate the present model.

## 2. The Governing Equations and the Simplification

When focusing just on a local region, water flows on the planet earth's surface can be considered as a phenomenon governed by the Navier-Stokes Equations.

$$\frac{\partial u}{\partial x} + \frac{\partial v}{\partial y} + \frac{\partial w}{\partial z} = 0 \tag{1}$$

$$\frac{\partial u}{\partial t} + u\frac{\partial u}{\partial x} + v\frac{\partial u}{\partial y} + w\frac{\partial u}{\partial z} = -\frac{1}{\rho}\frac{\partial p}{\partial x} + \frac{\mu}{\rho}\left(\frac{\partial^2 u}{\partial x^2} + \frac{\partial^2 u}{\partial y^2} + \frac{\partial^2 u}{\partial z^2}\right) \tag{2}$$

$$\frac{\partial v}{\partial t} + u\frac{\partial v}{\partial x} + v\frac{\partial v}{\partial y} + w\frac{\partial v}{\partial z} = -\frac{1}{\rho}\frac{\partial p}{\partial y} + \frac{\mu}{\rho}\left(\frac{\partial^2 v}{\partial x^2} + \frac{\partial^2 v}{\partial y^2} + \frac{\partial^2 v}{\partial z^2}\right) \tag{3}$$

$$\frac{\partial w}{\partial t} + u\frac{\partial w}{\partial x} + v\frac{\partial w}{\partial y} + w\frac{\partial w}{\partial z} = -g - \frac{1}{\rho}\frac{\partial p}{\partial z} + \frac{\mu}{\rho}\left(\frac{\partial^2 w}{\partial x^2} + \frac{\partial^2 w}{\partial y^2} + \frac{\partial^2 w}{\partial z^2}\right) \tag{4}$$

in which $u$, $v$, $w$ are components of the flow velocity vector in the $x$, $y$ and $z$ directions, $t$ is the time, $\rho$ is the density of water, $p$ is the gauge pressure in the water, $g$ is the gravitational acceleration, and $\mu$ is the coefficient of viscosity of the water. The $x$ and $y$ coordinates denote the horizontal directions while the $z$ coordinate denotes the vertical direction. The water flows are confined in the range of $z_b \leq z \leq \zeta$ in which $z_b(x,y)$ is the bathymetry while $\zeta(x,y,t)$ is the water surface elevation. The water depth $H$ is therefore defined as $\zeta - z_b$. Usually, the mean sea level is set at $z = 0$, so $z_b$ is negative in many cases, therefore, $h = -z_b$ is defined as the static water depth. Noted that is sometimes the static water depth just called the water depth and one should pay attention on the difference between $H$ and $h$.

The water pressure $p$ can be divided into two parts, the hydrostatic pressure and the non-hydrostatic pressure.

$$p = \rho g(\zeta - z) + q \tag{5}$$

in which $q$ is the non-hydrostatic pressure, also called the hydrodynamic pressure. The value of $q$ depends on the vertical position. On the free surface, it is zero. Following [15], the non-hydrostatic pressure at the bottom is symbolized as $Q$, and the gauge pressure at the bottom is expressed as

$$p = \rho g H + Q, \text{ at } z = z_b \tag{6}$$

The kinematic boundary condition on the free surface is expressed by the definition of the flow velocity.

$$w = \frac{D\zeta}{Dt} = \frac{\partial \zeta}{\partial t} + u\frac{\partial \zeta}{\partial x} + v\frac{\partial \zeta}{\partial y}, \text{ at } z = \zeta \tag{7}$$

At the bottom, though in reality the condition should be no-slip because of water viscosity, water there is regarded as capable to slide along the bottom surface in shallow water simulations. That is because in most practical applications the boundary layer near the bottom is relatively thin compared to the water depth. Consequently, the kinematic bottom boundary condition can be formulated as

$$w = \frac{Dz_b}{Dt} = u\frac{\partial z_b}{\partial x} + v\frac{\partial z_b}{\partial y}, \text{ at } z = z_b \tag{8}$$

The effect of bottom friction will be added in the model by employing the Manning's coefficient. The Keller-box scheme [12] was employed [15] to obtain the non-hydrostatic shallow water equations.

$$\frac{\partial \zeta}{\partial t} + \frac{\partial (UH)}{\partial x} + \frac{\partial (VH)}{\partial y} = 0 \tag{9}$$

$$\frac{\partial U}{\partial t} + U\frac{\partial U}{\partial x} + V\frac{\partial U}{\partial y} = -g\frac{\partial \zeta}{\partial x} - n_m^2 \frac{g}{H^{\frac{1}{3}}} \frac{U\sqrt{U^2+V^2}}{\rho H} - \frac{Q}{2\rho H}\left(\frac{\partial \zeta}{\partial x} + \frac{\partial z_b}{\partial x}\right) - \frac{1}{2\rho}\frac{\partial Q}{\partial x} \tag{10}$$

$$\frac{\partial V}{\partial t} + U\frac{\partial V}{\partial x} + V\frac{\partial V}{\partial y} = -g\frac{\partial \zeta}{\partial y} - n_m^2 \frac{g}{H^{\frac{1}{3}}} \frac{V\sqrt{U^2+V^2}}{\rho H} - \frac{Q}{2\rho H}\left(\frac{\partial \zeta}{\partial y} + \frac{\partial z_b}{\partial y}\right) - \frac{1}{2\rho}\frac{\partial Q}{\partial y} \tag{11}$$

$$\frac{\partial W}{\partial t} = \frac{Q}{\rho H} \tag{12}$$

in which $U$, $V$ and $W$ are the average values of $u$, $v$ and $w$ by the integration in the $z$ direction while $n_m$ is the Manning's coefficient. For finding the values of $Q$ at discretized nodes, or in the elements, solving a time independent Poisson equation is usually employed in most of the already published non-hydrostatic models [2,3,11,15,17,19]. This will form a huge global matrix system when the computational domain is discretized with a great number of nodes. Unlike those published non-hydrostatic models, the hydrodynamic pressure is obtained straightforwardly rather than from the solution of a time independent Poisson equation.

Following the assumption in [13,18] in which $W$ was treated as a linear distributed function in the $z$ direction, the value of $W$ can be formulated as

$$W = \frac{1}{2}\left(\frac{\partial \zeta}{\partial t} + U\left(\frac{\partial \zeta}{\partial x} + \frac{\partial z_b}{\partial x}\right) + V\left(\frac{\partial \zeta}{\partial y} + \frac{\partial z_b}{\partial y}\right)\right) \tag{13}$$

Applying Equations (9)–(13), we can obtain

$$W = \frac{1}{2}\left(U\left(\frac{\partial \zeta}{\partial x} + \frac{\partial z_b}{\partial x}\right) - \frac{\partial (UH)}{\partial x} + V\left(\frac{\partial \zeta}{\partial y} + \frac{\partial z_b}{\partial y}\right) - \frac{\partial (VH)}{\partial y}\right) \tag{14}$$

And Equation (12) can be written as

$$Q = \rho H \frac{\partial W}{\partial t} \tag{15}$$

So far we have a set of non-hydrostatic shallow water equations including Equations (9)–(11), (14) and (15) which are straight forwardly employed in the explicit time marching processes.

## 3. The Time Marching Processes

In this study, an explicit method is proposed. The method includes the prediction and the correction processes. Though the non-hydrostatic shallow water equations are two dimensional, due to the focus of this study is on the explicit time marching procedure used in water wave simulations, only one dimensional formulation is demonstrated to elucidate the time marching processes. All terms relevant to the $y$ direction are omitted.

The time domain is discretized with a small time increment $\Delta t$ whose size is determined by the consideration of numerical stability. At each time step, a prediction as well as several iterations of correction are processed.

### 3.1. Prediction

At the stage of prediction, terms relevant to $Q$ are provisionally omitted because the values of $Q$ and its partial derivatives at the next time step are unknown yet. The formulae for the prediction are obtained by the concept of the forward difference.

$$\zeta^{(*)} = \zeta^{(n)} + \Delta t\, S_c^{(n)} \tag{16}$$

$$U^{(*)} = U^{(n)} + \Delta t (S_{x1} + S_{x2} + S_{x3})^{(n)} \tag{17}$$

in which symbols with superscript $(n)$ are the physical quantities at the $n$-th time step and those with superscript $(*)$ are the provisional values of the relevant physical quantities at the $(n + 1)$-th time step. A subscript $c$ or $x$ bestowed on each symbol $S$ indicates that it is formulated as a source term relating to the mass conservation equation or the $x$ directional momentum equation. They are listed as follows.

$$S_c = -\frac{\partial(UH)}{\partial x} \tag{18}$$

$$S_{x1} = -U\frac{\partial U}{\partial x} \tag{19}$$

$$S_{x2} = -g\frac{\partial \zeta}{\partial x} \tag{20}$$

$$S_{x3} = -n_m^2 \frac{g}{H^{\frac{1}{3}}} \frac{U|U|}{\rho H} \tag{21}$$

The provisional values of $\zeta$ and $U$ can be obtained by using Equations (16) and (17). Then, one can calculate the provisional value of $W$

$$W^{(*)} = \frac{1}{2}\left(U\left(\frac{\partial \zeta}{\partial x} + \frac{\partial z_b}{\partial x}\right) - \frac{\partial(UH)}{\partial x}\right)^{(*)} \tag{22}$$

as well as the representative mean value of $Q$ in the time interval.

$$\overline{Q} = \rho \frac{H^{(n)} + H^{(*)}}{2} \frac{W^{(*)} - W^{(n)}}{\Delta t} \tag{23}$$

in which the over bar denotes the average value. So far we have the provisional values of $\zeta$, $U$ and $W$. They are regarded as true values at the $(n + 1)$-th time step.

### 3.2. Correction

The formulae for the correction are obtained by the concept of the Crank-Nicolson scheme.

$$\zeta^{(**)} = \zeta^{(n)} + \Delta t\, \overline{S_c} \tag{24}$$

$$U^{(**)} = U^{(n)} + \Delta t\left(\overline{S_{x1}} + \overline{S_{x2}} + \overline{S_{x3}} + r\left(\overline{S_{x4}} + \overline{S_{x5}}\right)\right) \tag{25}$$

in which the superscript $(**)$ denotes the correction, the over bar is defined as afore mentioned. It should be noted that for avoiding numerical blowup, a ramping treatment is introduced in which $r$ is the ramping factor. In this study, we use 5 iterations of correction. Value of $r$ is based on Equation (26). Its value is 1 for the last two iterations of correction

so the formulation consists with the Crank-Nicolson scheme when the entire prediction-corrector procedure is completed.

$$r = 1 - \cos^2\left(\frac{\pi}{2}\min\{1,\ i^*/4\}\right) \tag{26}$$

which indicates the factor of ramping treatment that the $i^*$-th iteration of correction is being processed. Here we have new source terms in the formulation due to the presence of $Q$. They are

$$S_{x4} = -\frac{Q}{2\rho H}\left(\frac{\partial \zeta}{\partial x} + \frac{\partial z_b}{\partial x}\right) \tag{27}$$

$$S_{x5} = -\frac{1}{2\rho}\frac{\partial Q}{\partial x} \tag{28}$$

The way of calculating the mean value in the time interval is the average of the previous time step and the provisional value of the processed time step. For example,

$$\overline{S_c} = \frac{S_c^{(*)} + S_c^{(n)}}{2},\ \text{or } \overline{S_{x5}} = -\frac{1}{2\rho}\frac{\partial \overline{Q}}{\partial x} \tag{29}$$

It should be noted that after each iteration of correction, all the provisional values of $\zeta$, $U$ and $W$ should be updated with those obtained by the correction. After 5 iterations of correction, the resulted values of $\zeta$, $U$ and $W$ are regarded as the converged values of $\zeta$, $U$ and $W$ at the $(n + 1)$-th time step.

## 4. Approach for Calculating the Spatial Derivatives

The weighted-least-squares local polynomial approximation is employed for calculating the spatial derivatives of the physical quantities. The formulation in this study is one dimensional, but one should keep in mind that its application can be easily extended to two dimensional problems. The computational domain is discretized with $N$ nodes. The numbering is free from the positions of the nodes. Arranging the nodes in an equal spacing is not necessary. These are merits of the meshless methods. Taking $\zeta$ as an example, at an specific node locating at $x_j$, the free surface elevation around that node can be approximated by using the second degree local polynomial

$$\zeta_{x \approx x_j} \approx \hat{\zeta}_j(x) = \sum_{i=1}^{3} \alpha_{ji} p_{ji} \tag{30}$$

in which $p_{j1} = 1$, $p_{j2} = x - x_j$, and $p_{j3} = (x - x_j)^2/2$. One may use higher degree local polynomial for accuracy. Once the coefficients of this local approximation are determined, the first and the second order derivatives at $x = x_j$ can be approximated as $\alpha_{j2}$ and $\alpha_{j3}$. For determining these coefficients, values of $\zeta$ at neighbor nodes of $x_j$ are needed. If just two neighbor nodes are used, it is the finite difference method. In this study, more than two neighbor nodes are included because the method used here is a meshless one. Applying values of $x$ at all the nodes to Equation (30) and introducing the weighting factors, the error residual of this local approximation is defined as

$$E_j = \sum_{l=1}^{N} W_{jl}\left(\zeta(x_l) - \hat{\zeta}_j(x_l)\right)^2 \tag{31}$$

in which $W_{jl}$ is the weighting factor whose value is between 0 and 1, and is dependent on the distance from $x_j$ to $x_l$ (i.e., $r_{jl} = |x_l - x_j|$). In this study, the monotonously decaying function is used for determine the value of $W_{jl}$.

$$W_{jl} = \begin{cases} (1 - r_{jl}/\rho_j)^\varepsilon & ,\ r_{jl} < \rho_j \\ 0 & ,\ r_{jl} \geq \rho_j \end{cases} \tag{32}$$

where $\varepsilon$ is the shape parameter and $\rho_j$ denotes the size of the supporting range around the $j$-th node. One may choose other functions for this purpose. Although $W_{jl}$ is determined by a radial basis function, it is treated as a "factor" in the process of seeking the partial derivatives of $\zeta$. By skipping the zero-valued terms and using $k$ as the local index of the node at $x_l$, Equation (31) can be rewritten as

$$E_j = \sum_{k=1}^{n_{loc}} W_{j\underline{k}} \big(\zeta(x_{\underline{k}}) - \hat{\zeta}_j(x_{\underline{k}})\big)^2 \tag{33}$$

in which $k$ represents the local index of the $l$-th node if it is inside the $j$-th supporting range. The underline is to emphasize it is a local index. The symbol $n_{loc}$ denotes the number of nodes enclosed in the range of $|x - x_j| < \rho_j$. A algebraic system of linear equations is formed by the values of $\zeta$ at all the nodes inside the supporting range.

$$\mathbf{A}\boldsymbol{\alpha}_j = \boldsymbol{\beta} \tag{34}$$

in which

$$\mathbf{A} = \begin{bmatrix} a_{11} & a_{12} & a_{13} \\ a_{21} & \vdots & \vdots \\ \vdots & \vdots & \vdots \\ a_{n_{loc}1} & a_{n_{loc}2} & a_{n_{loc}3} \end{bmatrix} \tag{35}$$

$$\boldsymbol{\alpha}_j = \begin{bmatrix} \alpha_{j1} & \alpha_{j2} & \alpha_{j3} \end{bmatrix}^T \tag{36}$$

$$\begin{aligned} \boldsymbol{\beta} &= \begin{bmatrix} \beta_1 & \cdots & \beta_k & \cdots & \beta_{n_{loc}} \end{bmatrix}^T \\ &= \begin{bmatrix} w_1\zeta_{\underline{1}} & \cdots & w_k\zeta_{\underline{k}} & \cdots & w_{n_{loc}}\zeta_{\underline{n_{loc}}} \end{bmatrix}^T \end{aligned} \tag{37}$$

where $w_k = \sqrt{W_{j\underline{k}}}$, $\zeta_{\underline{k}} = \zeta(x_{\underline{k}})$, and $a_{ki} = w_k p_{ji}(x_{\underline{k}})$. The underlines in Equation (37) remind the relation between the local sequential number and the global sequential number. We keep the subscript in $\boldsymbol{\alpha}_j$ to remind this local approximation is only valid in the vicinity of $x_j$. Once we move our focus to another node, there will be a new set of $\alpha_{ji}$, $n_{loc}$, and a new matrix $\mathbf{A}$. Values of $\alpha_{j1}$, $\alpha_{j2}$ and $\alpha_{j3}$ are obtained by the least-squares approach. We use the same way to calculating the spatial derivatives of other physical quantities such as $U$, $W$ and $Q$. Because the weighting factors are introduced in Equation (34), it is so called the weighted-least-squares (WLS) approach. The purpose of the weighting factor is to reduce the error at the focused point and to diminish the differences between the approximated and the exact values. The faster the weighting factor decays by the distance from the focused point, the closer approximation to the exact value at the focused point will be. This means larger $\varepsilon$ in Equation (32) results in closer approximation, but this could increase numerical instability, especially when the time marching scheme employed is an explicit one. In this study, a smoothing process is suggested in addition to carefully choosing the value of $\varepsilon$. The suitable value of shape parameter $\varepsilon$ is tested and found related to the water depth, nodal resolution, and the time increment. It will be furtherly discussed in the first example case. In all the numerical simulations of this study, we set $n_{loc}$ as 15, so each local approximation needs at least 14 neighbor nodes. The value of $\rho_j$ is determined as multiplying the distance to the 14th nearest neighbor node by 1.01.

The weighted-least-squares approach is much similar to the moving-least-squares (MLS) approach. The difference is, in the WLS, the weightings are just factors, but in the MLS, the spatial derivatives of these weightings take important places in calculating the besought approximations. The weightings are also used as the smoothing kernel function in some relevant numerical methods such as the Smooth Particle Hydrodynamics [31–33].

## 5. The Smoothing Process

As mentioned previously, an additional smoothing process is suggested to enhance the numerical stability. It is employed on smoothing the values of $\overline{Q}$ in which the overbar indicates the average in the time interval. Equation (38) is used in the smoothing process

$$\overline{Q}_j^{(s)} = \sum_{k=1}^{n_{loc}} W_{j\underline{k}}\overline{Q}_{\underline{k}} / \sum_{k=1}^{n_{loc}} W_{j\underline{k}} \tag{38}$$

in which the superscript $(s)$ is used to indicate a smoothed value, $W_{j\underline{k}}$ is the weighting factor afore mentioned, subscript $j$ denotes the sequential number of a specific node while subscript $k$ denotes the local index of a node in this supporting range and the underline emphasizes the relation between the local sequential number and global sequential number. After all the smoothed values are calculated, we use them to replace the original values of $\overline{Q}$ calculated by Equation (23).

## 6. Examples

Four example cases are tested in the present study, including the propagation of a solitary wave in a constant depth, a solitary wave reflection after running up a slope, nonlinear waves generated by the periodic motion of a piston-type wave maker, and the nonlinear modulation of periodic waves passing over a submerged obstacle.

### 6.1. Propagation of a Solitary Wave in a Constant Depth

Solitary wave propagation in a one-dimensional frictionless channel with a constant depth is used as the benchmark for the model verification. It is also used to test the shape parameter of the function for determining the weighting factor. The solitary wave for the shallow water equations is

$$\zeta(x,t) = A\,\text{sech}^2\left(\sqrt{\frac{3A}{4h^3}}(x - x_0 - ct)\right) \tag{39}$$

in which $A$ is the amplitude, $x_0$ is the initial position of the wave crest, and $c = \sqrt{g(A+h)}$ is the propagation speed of the wave, and

$$U(x,t) = \frac{c\zeta}{\zeta + h} \tag{40}$$

$$W(x,t) = -\frac{H}{2}\frac{\partial U}{\partial x} \tag{41}$$

It should be noted that Equation (41) is based on the assumption that $w$ is linear distributed in the vertical direction. The water depth $h$ chosen is 10 m, while the amplitude $A$ is 2 m. The length of the numerical channel is 1000 m. The initial position of the wave crest is at $x = 200$ m. The domain is discretized with constant nodal spacing $\Delta x$ from 0.5 m to 3 m in various tests. It is presumed that the suitable value of shape parameter $\varepsilon$ in Equation (32) is related to $C_r$ and $\Delta x/h$, in which $C_r = \sqrt{gh}/(\Delta x/\Delta t)$ is the computational Courant number. Totally 89 runs with various time increment $\Delta t$ from 0.0008 s to 0.04 s are conducted. The range of Courant number $C_r$ is from 0.0026 to 0.132. In each run, the value of $\varepsilon$ is tuned until the root mean square error of $\zeta$ at $t = 50$ s is minimized. The root mean square error is defined as

$$E_{rms} = \sqrt{\sum_{j=1}^{N}\left(\zeta_j - \zeta_j^{(e)}\right)^2 / N} \tag{42}$$

in which $\zeta_j$ is the value of $\zeta$ at the $j$-th node and the superscript $(e)$ indicates the "exact solution". After all the runs are performed, a function $\varepsilon = \varepsilon(C_r, \Delta x/h)$ is obtained by the multiple value regression. In this study, the third degree polynomial is chosen as

the regressive function. In all the further example cases, the value of $\varepsilon$ is determined by this regressive function. Figure 1 shows the results of $(\Delta x, \Delta t) = (0.5 \text{ m}, 0.001 \text{ s})$ and $(\Delta x, \Delta t) = (2.5 \text{ m}, 0.02 \text{ s})$. The root mean square errors are 0.0155 m and 0.0345 m, respectively. The exact solution and the numerical result of [3] are also plotted for comparison. The comparison shown in Figure 1 indicates that accuracy is improved using smaller nodal resolution and time increment. The solitary wave in the numerical model of [3] seems propagating faster slightly. As we look at the position of the wave crest, the result of the present model is closer to the exact solution.

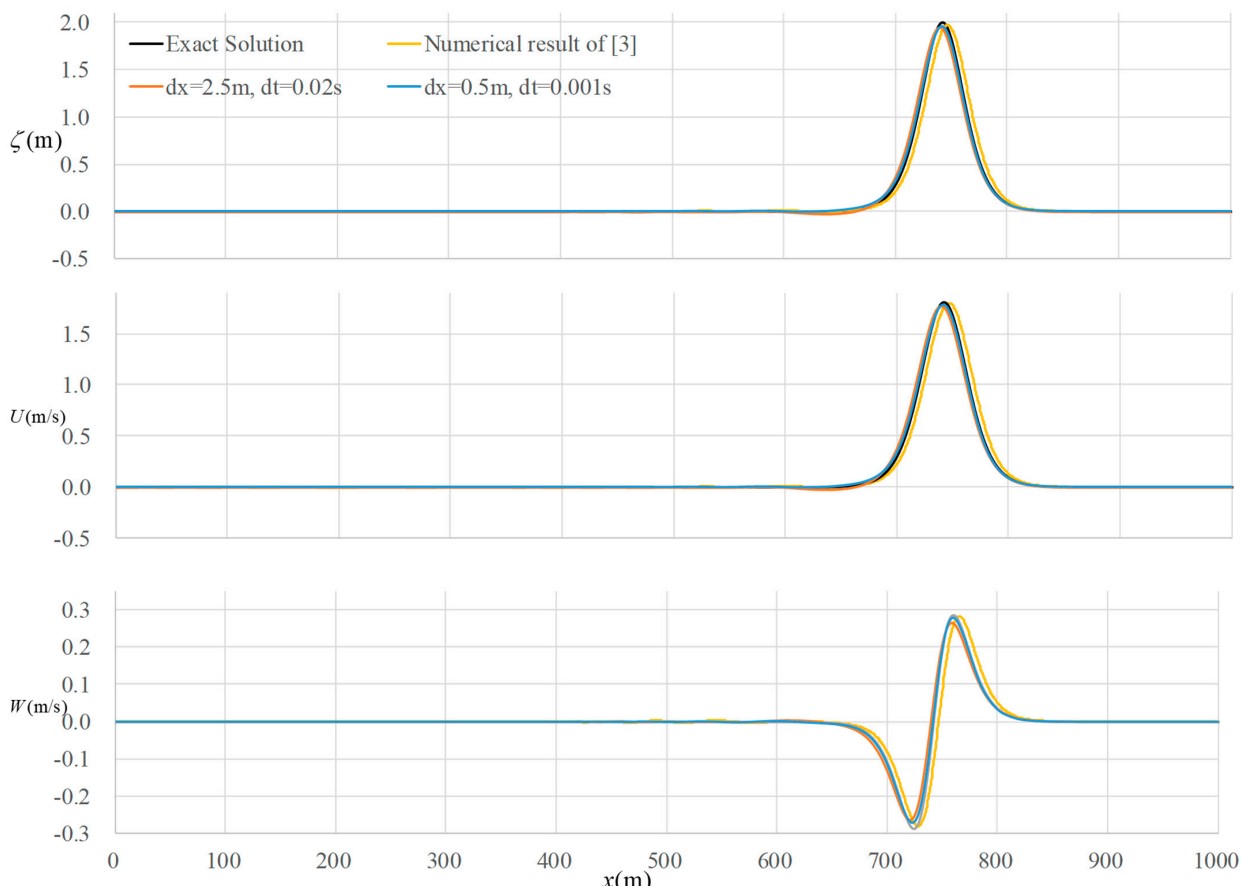

**Figure 1.** Comparison of the present numerical results with the exact solution and other numerical results of a solitary wave propagation in a constant depth.

### 6.2. Solitary Wave Reflection after Running Up a Slope

This test case was reported in [34,35] which said the experiment data are available in [36]. The bathymetry of this test case is shown in Figure 2.

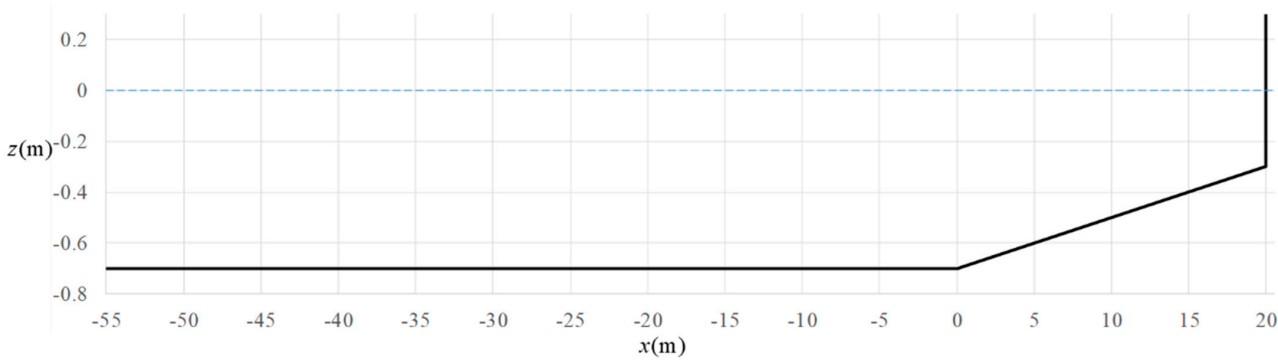

**Figure 2.** Bathymetry for test case of solitary wave reflection after running up a slope.

The domain is 75 m in length which includes a constant depth region and a 1:50 slope. At the end of the flume a vertical wall that can reflect waves is placed. The water depth is 0.7 m in the plain region and is 0.3 m at the end wall. A solitary wave which is initially centered at $x = -30$ m is used as the incidence. The amplitude of the solitary wave is 0.07 m. Nodes in the domain are arranged with various nodal resolution according to the water depth. The nodal spacing is 0.125 m in $-55$ m $\leq x \leq 13$ m, 0.1 m in 13 m $\leq x \leq 16$ m, and 0.08333 m in 16 m $\leq x \leq 20$ m, respectively. The time increment $\Delta t$ is chosen as 0.005 s, which represents the maximal Courant number $C_r$ of 0.116. The processes from forward propagation, shoaling, reflection, and backward propagation within 0 s $\leq t \leq 30$ s is simulated. Snapshots of the free surface profiles at $t = 14$ s, 15 s, ..., 25 s are shown in Figure 3. In $t = 14$–17 s one can see the wave becomes more and more inclined towards the shallower water region which represents the shoaling process. At $t = 18$ s one can see the water level gets higher than the static water level which means the reflection has begun. The water level at the vertical wall reaches the highest point at about $t = 19$ s. After that, the reflecting waves begin the backward propagation. Noted that there is only one incident wave and at least two reflecting wave are generated.

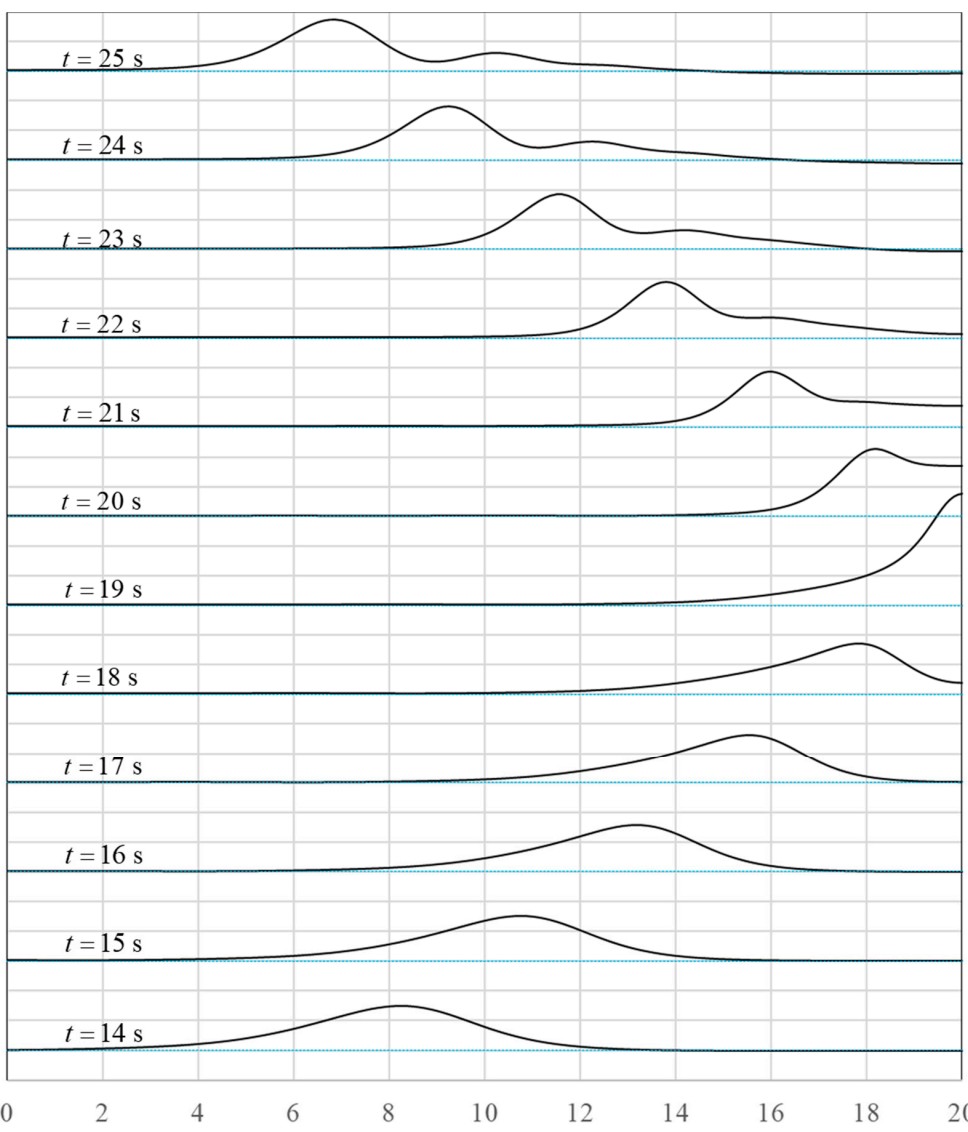

**Figure 3.** Snapshots of the free surface profiles in the test case of solitary wave reflection after running up a slope. Black solid lines are the free surface. The blue dash lines are the static water level. The height of each cell is 0.05 m.

Numerical results are compared with the observed time series of the free surface elevation at $x = 0$ m, $x = 16.25$ m, and $x = 17.75$ m, respectively, shown in Figure 4. The numerical results of [35] are also potted in this figure for comparison. On each panel the first peak corresponds to the incident wave while the second and subsequent peaks are referred to the reflected waves. The comparison shows that the prediction of the present model is quite close to the numerical result of [35] which was computed with a finite element model.

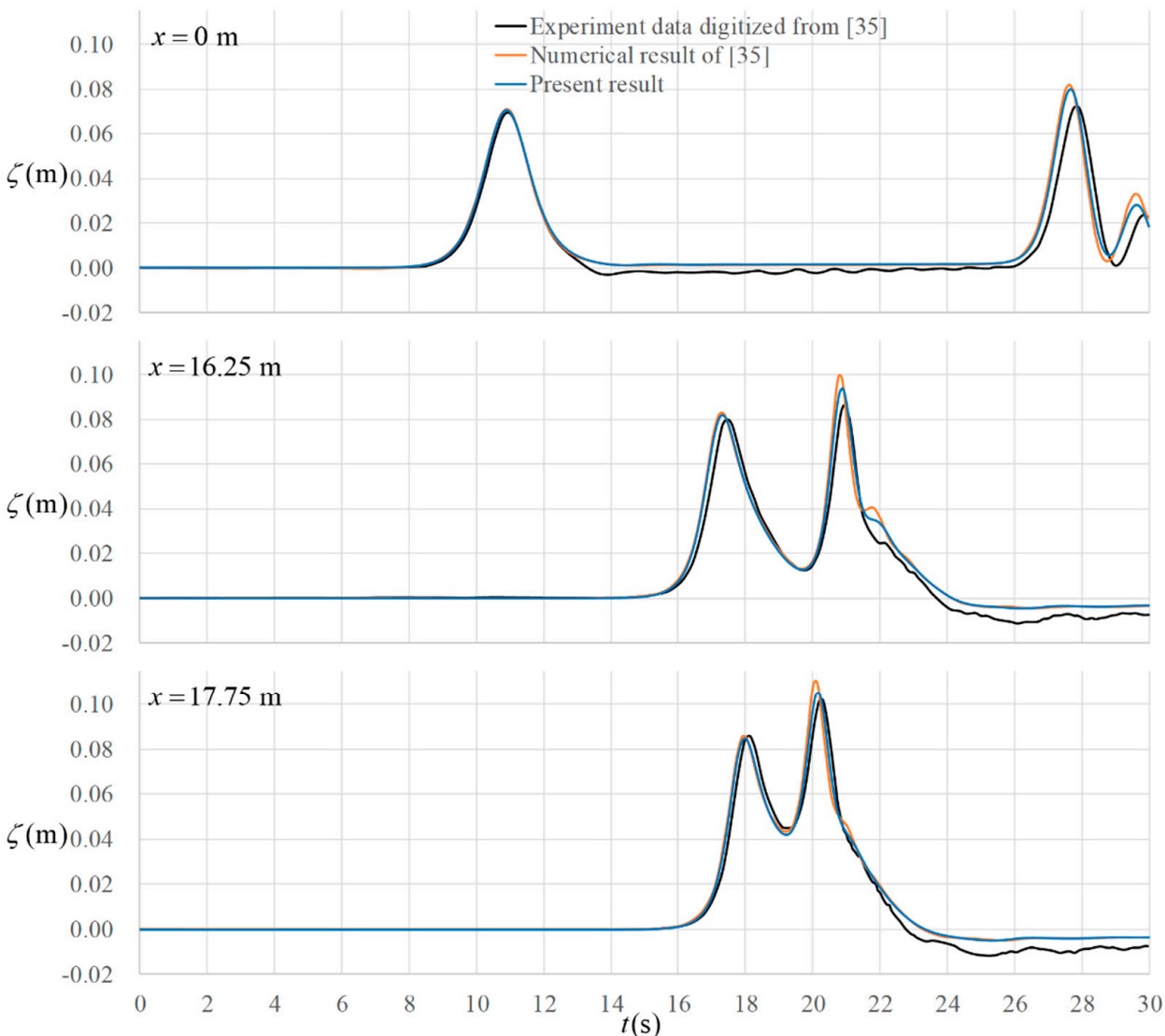

**Figure 4.** Comparison of the present numerical results with the other data for the case of solitary wave reflection after running up a slope.

### 6.3. Nonlinear Waves Generated by a Large Stroke Harmonic Motion of a Piston Type Wave Maker

According to the wave maker theory [37], harmonic motion of a piston type wave maker generates sinusoidal waves on the water surface. However, when the stroke of the wave paddle is large enough, nonlinear effect shows up and the generated waves will be more or less irregular. This was reported in [38] and validated experimentally. The water depth in this case is 0.38 m and wave period is 2.75 s. The stroke of the wave paddle is 12.2 cm. Consequently, the boundary condition at $x = 0$ is given by

$$U_0(t) = 0.13937 \sin(2.2848t) \tag{43}$$

The unit of $U_0$ is m/s. The domain of computation is 50 m in length. Simulation of 0 s $\leq t \leq$ 22 s is performed. Numerical results are compared with the observed time series of the free surface elevation at $x = 4.9$ m and $x = 8.7$ m, as well as with the analytic solution [38] and other numerical data [39]. Figure 5 shows the comparisons. In [38,39], only the peak-to-peak data within one wave period were shown. We match our results with the experiment data at the peaks and check the phase difference between the two wave gauges. At $x = 4.9$ m, the comparison starts at $t = 14.22$ s. According to the dispersive relation, the wavelength of the prime constituent is 5.13 m. The distance between the two wave gauges is 3.8 m. Therefore, the shift of the peaks between the two wave gauges is 0.74 s. This is validated in Figure 5. The comparison shows that the numerical result is quite close to the experiment data and the analytical result [38]. Figure 6 shows the comparison of the free surface profile at $t = 21.8$ s with the result of [39]. Good agreement between the two numerical results through the entire domain is found.

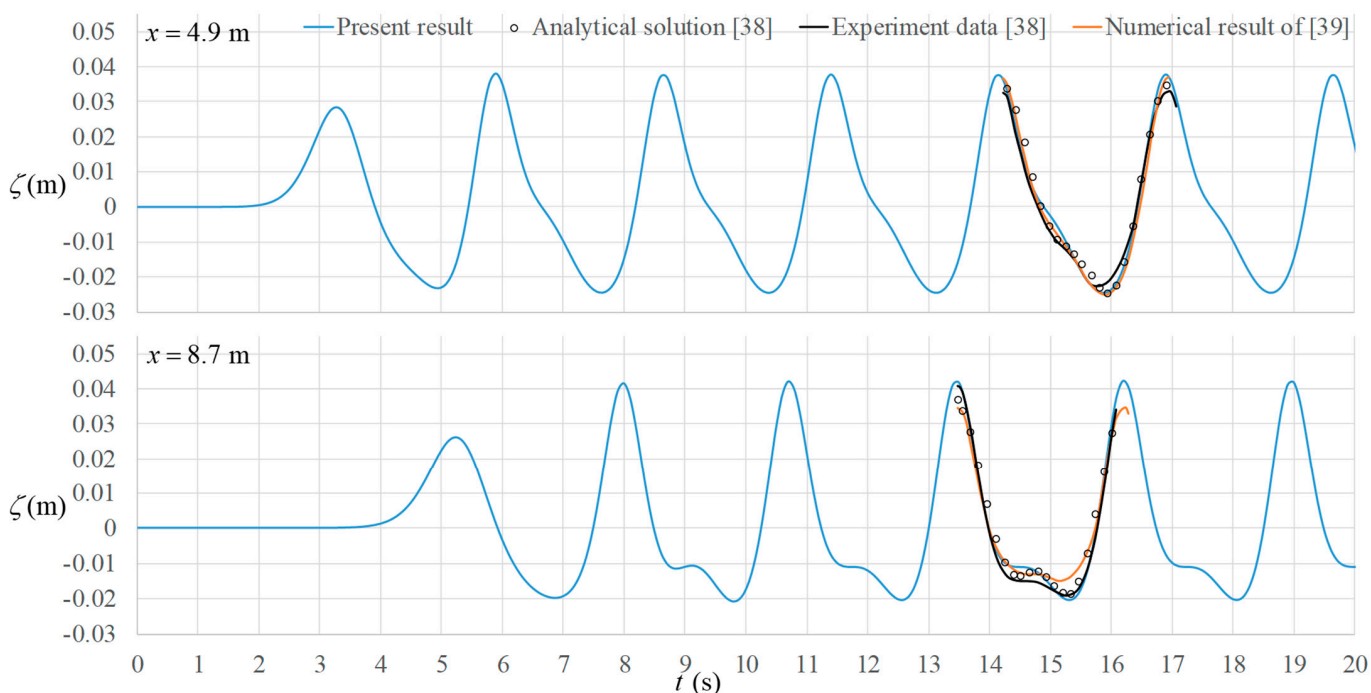

**Figure 5.** Comparison of the present numerical results with the other data (experiment data [38], analytic solution [38], and the numerical results of [39]) at wave gauges for the case of nonlinear waves generated by a large stroke harmonic motion of a piston type wave maker.

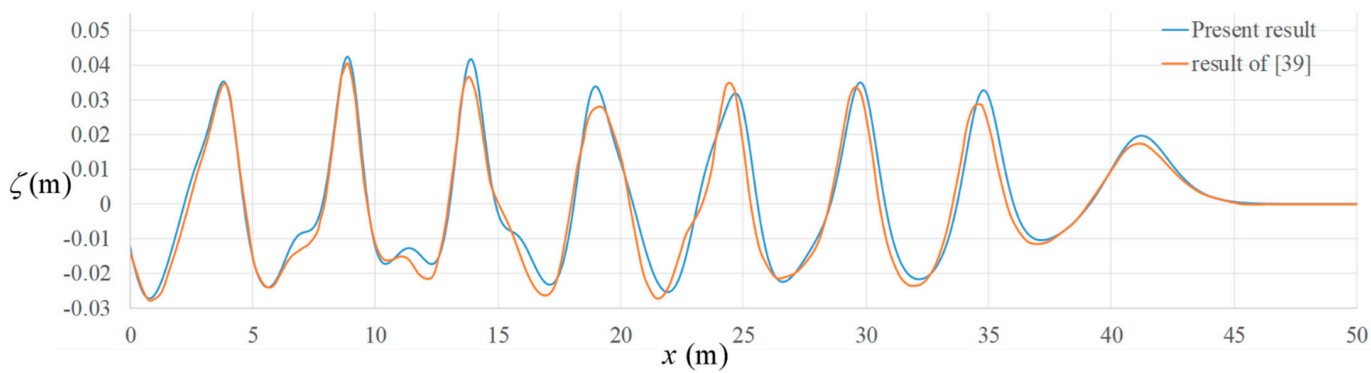

**Figure 6.** Comparison of the present numerical results with the numerical result of [39] on the free surface profile at $t = 21.8$ s for the case of nonlinear waves generated by a large stroke harmonic motion of a piston type wave maker.

*6.4. The Nonlinear Modulation of Periodic Waves Passing over a Submerged Obstacle*

Several non-breaking wave tests in a physical water flume were carried out by [40] to study the modulation of the monochromatic waves traveling over a submerged structure and were used to verify their numerical boundary element method (BEM) model. The layout of the experiment is depicted in Figure 7. The toe of the submerged dike is at $x = 6$ m. The water depth is 0.4 m in front of the obstacle and 0.1 m above the dike. The slopes of the front part and the rear part of the dike are 1:20 and 1:10, respectively. The width of the top is 2 m. In the experiment, a 1:25 slope starting at $x = 18.95$ m was placed in the rear part of the flume to dissipate the wave energy from reflection by wave breaking. We use a shallow water region with a very rough bottom as a replacement for wave energy dissipater. The length and the depth of this shallow water region are 2.5 m and 0.06 m, respectively. The bottom of the channel is frictionless in the range of 0 m $\leq x \leq$ 18.95 m and very rough in the range of $x > 18.95$ m. The Manning's coefficient in the rough bottom region is 0.02.

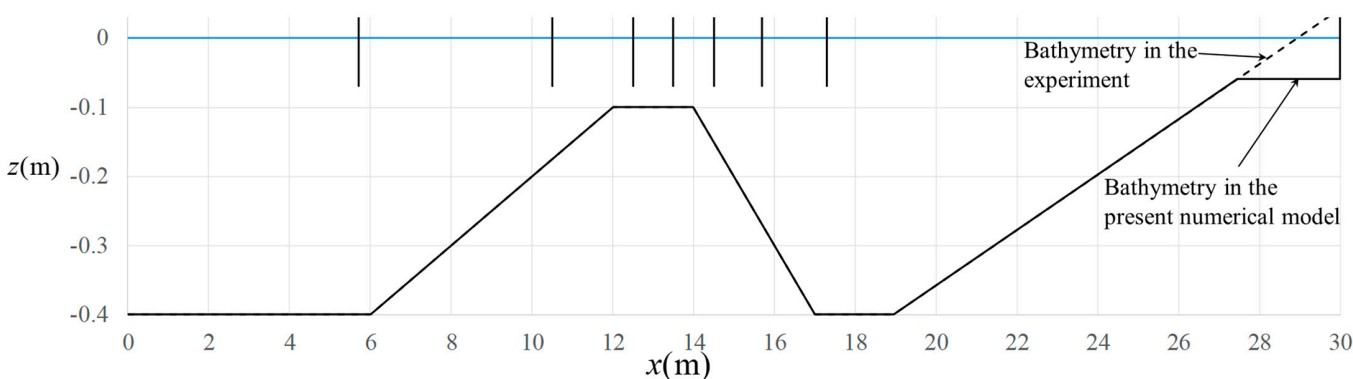

**Figure 7.** Layout of the experiment in [40] and the bathymetry of the present numerical model.

Time series of the free surface elevation observed at 7 wave gauges are used for comparison. The positions of these wave gauges are $x = 5.7$ m, $x = 10.5$ m, $x = 12.5$ m, $x = 13.5$ m, $x = 14.5$ m, $x = 15.7$ m, and $x = 17.3$ m, respectively. The case of incident wave height 0.02 m and the wave period 2 s is chosen for verification. According to the wave maker theory [37], the stroke of the wave paddle in the wave maker is 2.953 cm. Therefore, the boundary flow velocity given at $x = 0$ is

$$U_0(t) = 0.046389 \sin(3.1416t) \tag{44}$$

The unit of $U_0$ is m/s.

Nodes in the computational domain are irregularly distributed, with various nodal spacing in accordance with the water depth. Totally, there are 430 nodes. The nodal spacing is controlled by the condition that $h/\Delta x$ is greater than 1.5. The range of nodal spacing is 0.04–0.125 m, as shown in Figure 8.

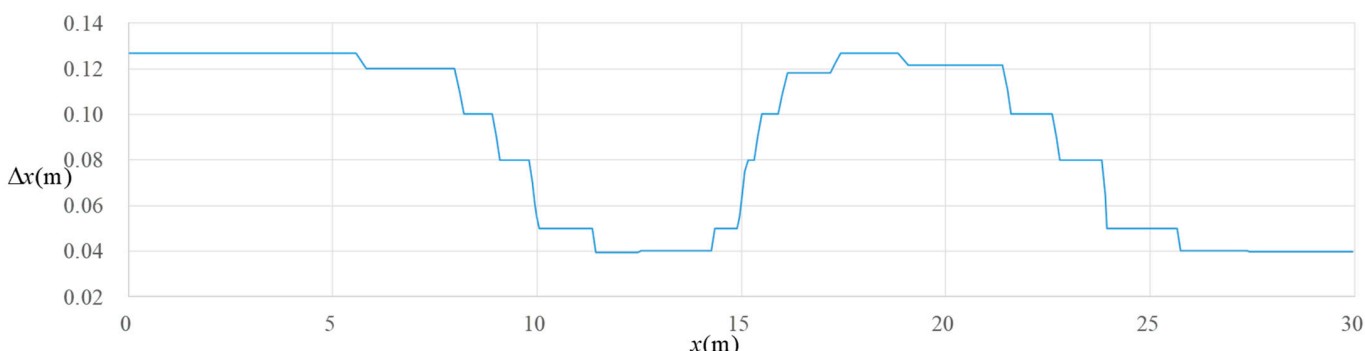

**Figure 8.** The nodal spacing for the simulation of the nonlinear modulation of periodic waves passing over a submerged obstacle.

The time increment is used as 0.005 s. Simulation of the flow in $0\,\text{s} \le t \le 30\,\text{s}$ is performed. Snapshots of the water surface profiles at $t = 28$ s and $t = 30$ s are plotted in Figure 9. The two profiles are almost identical to each other, note the wave period is 2 s. This means the rough bottom in $x > 18.95$ m effectively dissipates the wave energy from reflection. One can also see in this figure that in front of the submerged dike, the free surface waves are quite regular in a sinusoidal shape. As the waves propagate in the sloping region, they incline forwardly due to the shoaling effect. When they move through the top of the submerged dike and continue going into the region behind the submerged dike, higher harmonic components are generated and the waves become more or less irregular.

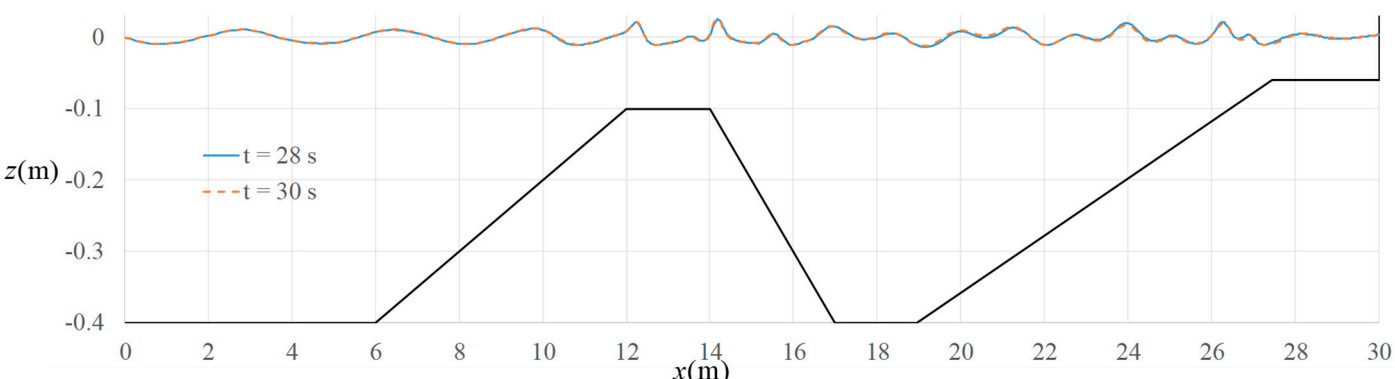

**Figure 9.** Snapshots of the water surface profiles at $t = 28$ s and $t = 30$ s in the simulation of nonlinear modulation of periodic waves passing over a submerged obstacle.

Time series of the free surface elevation at the 7 wave gauges are shown in Figure 10. The comparison shows that the performance of present non-hydrostatic SWE model works as well as that of the BEM model of [40]. The results are close to the experiment data except at the seventh wave gauge. However, the result there is still acceptable. It seems that higher order nonlinear components are more pronounced behind the submerged obstacle. Treating the velocity component $w$ as a linear distributed function in the $z$ direction in the present model could be the reason of this discrepancy.

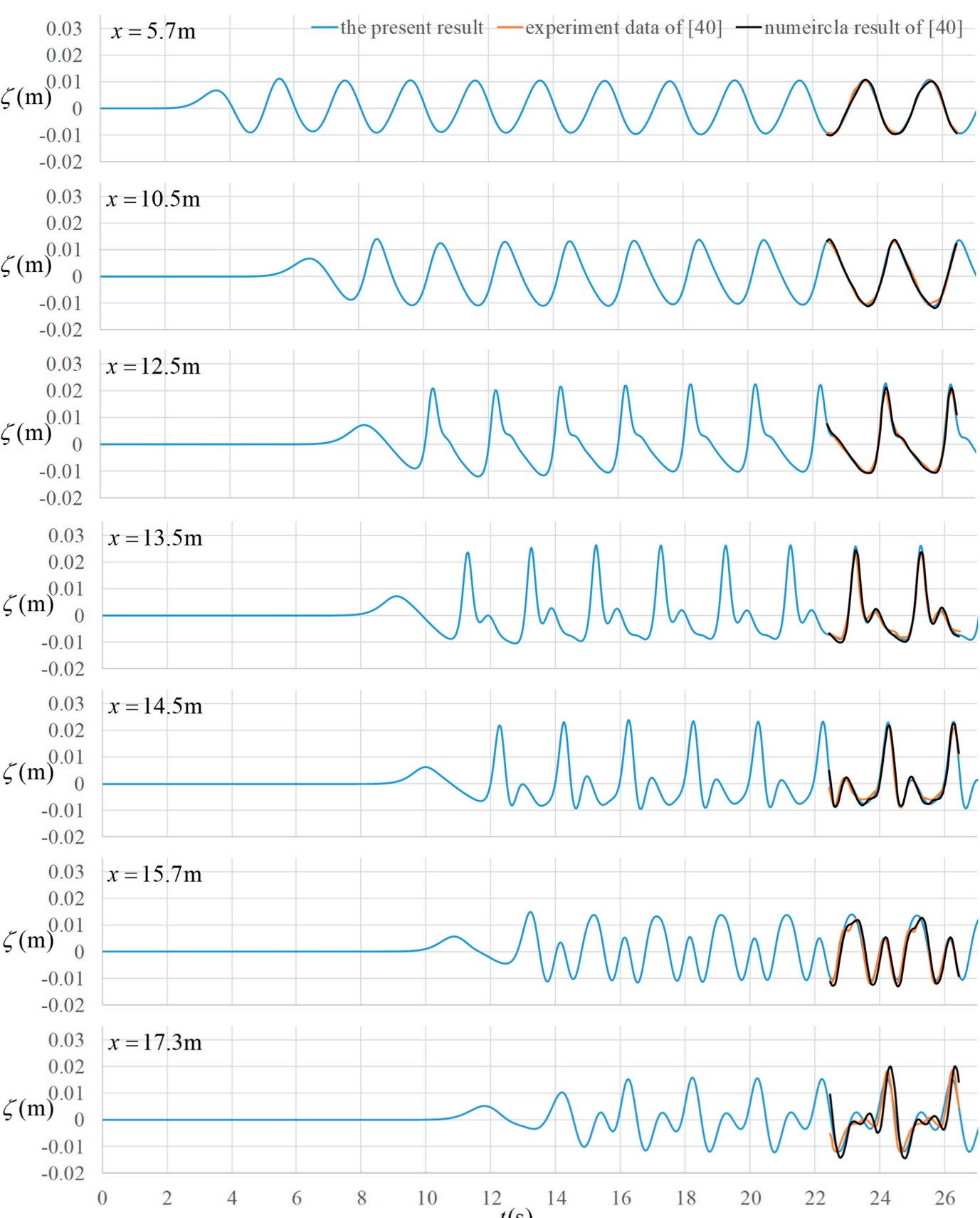

**Figure 10.** Comparison of the present numerical results with other data (numerical result of [40] and experiment data of [40]) at wave gauges for the case of nonlinear waves generated by a large stroke harmonic motion of a piston type wave maker.

## 7. Conclusions

An explicit time marching procedure for the non-hydrostatic shallow water equations with meshless approach is developed in this study. The meshless method with the local polynomial approximation and the weighted-least-squares (WLS) approach is employed to calculate the spatial derivatives of the physical quantities such as the temporal water depth, the average velocities in the horizontal and vertical directions, and the dynamic pressure at the bottom. Four benchmark problems are used to demonstrate the performance of the model.

In the first test case, we compare our results with the exact solution of the solitary wave propagation in a constant depth. The comparison shows that the numerical result is closer to the exact solution when smaller nodal spacing and time increment is used. This case is also used to tune the numerical parameters so the model can work properly. These numerical parameters are then used in further test cases to show that the present numerical model is stable and consistent.

The second case is the simulation of a solitary wave reflection after running up a slope. The processes from forward propagation, shoaling, reflection, and backward propagation are demonstrated. The results are compared with experiment data and other numerical results. Good agreement is found.

In the third case, nonlinear waves generated by a large-stroke harmonic motion of a piston type wave maker is simulated. Theoretically, harmonic motion of a piston type wave maker generates sinusoidal waves on the water surface. However, when the stroke of the wave maker is large, the generated waves can generate higher harmonic constituents, which are due to the nonlinear effects. The numerical results are validated by the comparison with the analytical solution, experiment data and other numerical results. Good agreements are found.

Finally, we simulate the nonlinear modulation of periodic waves passing over a submerged obstacle. In this case, regular waves are generated by a wave maker. These waves propagate forwardly and modulate when passing over a submerged obstacle. The results are compared with data collected at seven wave gauges. The time series of the water surface elevation at the first six wave gauges are almost identical to the experiment data. Though some slight discrepancy is found at the seventh wave gauge, the result is still quite acceptable. The reason of this slight discrepancy could be that we regard the velocity component $w$ linear distributed in the $z$ direction.

**Author Contributions:** Conceptualization, N.-J.W.; methodology, N.-J.W.; coding, N.-J.W.; investigation, Y.-M.S., S.-C.H., S.-J.L., T.-W.H.; data curation, Y.-M.S.; formal analysis, N.-J.W., Y.-M.S.; writing—original draft preparation, N.-J.W.; writing—review and editing, S.-C.H., S.-J.L., T.-W.H. All authors have read and agreed to the published version of the manuscript.

**Funding:** This research was funded by the Ministry of Science and Technology, Taiwan (Grant No. MOST 110-2221-E-019-026).

**Institutional Review Board Statement:** The study was conducted according to the guidelines of the Declaration of Helsinki, and approved by the Institutional Review Board.

**Informed Consent Statement:** Informed consent was obtained from all subjects involved in the study.

**Data Availability Statement:** All the data are available by digitizing the shown figures and can be traced back to references cited in this paper.

**Conflicts of Interest:** The authors declare no conflict of interest.

## Abbreviations

The following abbreviations are used in this manuscript:

| | |
|---|---|
| WLS | Weighted-least-squares |
| MLS | Moving-least-squares |
| SWE | Shallow water equations |
| FDM | Finite difference method |
| FEM | Finite element method |
| BEM | Boundary element method |

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
