# Peer review of "A Weighted-Least-Squares Meshless Model for Non-Hydrostatic Shallow Water Waves"

_water, doi:10.3390/w13223195_

Round 1
Reviewer 1 Report
I suggest the following modifications to make it acceptable for the journal: Water.
1. Specify the novelty of the current study.
2. Strong references are needed for governing equations (9)-(12).
3. Discussion of results may be improved by explaining in physical point of view.
4. What are the advantages of the method you used compare to other methods?
5. Some pictorial description of method may be added such as, the geometry discretization.
6. Paper must be checked for typos and grammar.
7. A fully contained nomenclature must be added to the text and all abbreviations must be explained.
8. Several typo errors are found and language improved.
Author Response
I suggest the following modifications to make it acceptable for the journal: Water.
- Specify the novelty of the current study.
Thank you very much for this suggestion. This is the first time a meshless method being used for solving the non-hydrostatic shallow water equations. We propose a truly explicit time marching procedure which needs not solve the Poisson equation for the dynamic pressure at the bottom. These two points are the novelties of the present study. They are stated in the introduction. Please see the highlighted in lines 46-49 and lines 52-55.
- Strong references are needed for governing equations (9)-(12).
These equations are derived by [15]. It is stated in the highlighted in lines 87-88.
- Discussion of results may be improved by explaining in physical point of view.
Thank you very much for this suggestion. Please see the highlighted in lines 260-266 for the second test case, lines 299-302 for the third test case, and lines 347-351 for the forth test case.
- What are the advantages of the method you used compare to other methods?
The advantage is this method is meshless. Though it is employed just in one dimensional problems, one can still find its merit. The numbering is free from the positions of the nodes. Arranging the nodes in an equal spacing is not necessary. Please see the highlighted in lines 150-151. It is more convenient than grid/mesh based methods.
- Some pictorial description of method may be added such as, the geometry discretization.
Thank you very much for this suggestion. The geometry discretization of the forth case is added as Figure 8. As per the geometry discretization of the first three cases, we think it is quite simple so we just describe them in texts.
- Paper must be checked for typos and grammar.
Thank you very much for this suggestion. Typos and grammar are checked carefully.
- A fully contained nomenclature must be added to the text and all abbreviations must be explained.
Thank you very much for this suggestion. Each abbreviation has been checked with its nomenclature. A list of abbreviation is also added in the manuscript, just before the list of references.
- Several typo errors are found and language improved.
Thank you very much for this suggestion. Typos and grammar are checked carefully.

Reviewer 2 Report
Review round 1
The manuscript is devoted to the development of a new scheme based on a Weighted-Least-Squares Meshless method for dealing with non-hydrostatic shallow water waves. Specifically, the Weighted-Least-Squares scheme is used to compute easily the spatial
derives of the of the physical quantities involved in the problem. The model and code is original and the manuscript has relatively suitable impact and relevance to the journal. All the results are well presented and support the correctness of the scheme as well as it applicability. I believe that after several minor revisions the manuscript can be considered for publication.
I would include a convergence analysis.
line 94 “This procedure consumes very much CPU time.” This sentence this sentence is a bit approximate. Could you reformulate it? Could you support the before mention statement with some proper reference?
Line 108: The sentence “The time domain is discretized with a small time increment” is not clear at all. What does it mean small? Respect to what? Can you provide a general rule for selecting the time step? Is the “Courant–Friedrichs–Lewy condition” a valid alternative?
Line 119: “With” should not have capital letter
It seems that all equation miss punctuation, see for example Eq. 29.
In the framework of meshless methods especially based on least-squares schemes, I would suggest to mention Moving-Least-Squares WENO schemes. These are indeed particularly useful for both spatial and temporal derivative reconstructions. See for example “Avesani et al, An alternative SPH formulation: ADER-WENO-SPH, Computer Methods in Applied Mechanics and Engineering”, “Antona et al., Towards a High Order Convergent ALE-SPH Scheme with Efficient WENO Spatial Reconstruction, Water, 2021”, “Avesani et al, An alternative smooth particle hydrodynamics formulation to simulate chemotaxis in porous media, Journal of mathematical biology, 2017”.
I would add proper labels and legends in the figure. I find difficult sometimes extrapolate all information from the captions of the figures sometimes.
Author Response
The manuscript is devoted to the development of a new scheme based on a Weighted-Least-Squares Meshless method for dealing with non-hydrostatic shallow water waves. Specifically, the Weighted-Least-Squares scheme is used to compute easily the spatial derives of the of the physical quantities involved in the problem. The model and code is original and the manuscript has relatively suitable impact and relevance to the journal. All the results are well presented and support the correctness of the scheme as well as it applicability. I believe that after several minor revisions the manuscript can be considered for publication.
We are grateful that the reviewer reads our manuscript carefully and gives us helpful suggestions.
I would include a convergence analysis.
line 94 “This procedure consumes very much CPU time.” This sentence this sentence is a bit approximate. Could you reformulate it? Could you support the before mention statement with some proper reference?
We said that because a huge global matrix system will be formed when the computational domain is discretized with a great number of nodes. Therefore, we prefer finding the values of Q directly and use the explicit method. This sentence has been rephrased. Please see the highlighted in lines 93-94.
Line 108: The sentence “The time domain is discretized with a small time increment” is not clear at all. What does it mean small? Respect to what? Can you provide a general rule for selecting the time step? Is the “Courant–Friedrichs–Lewy condition” a valid alternative?
Thank you very much for pointing out this. We rephrase this sentence as “The time domain is discretized with a small time increment whose size is determined by the consideration of numerical stability” to make it more clear. Please see the highlighted in lines 110-111.
Line 119: “With” should not have capital letter
This sentence has been rephrased as “The provisional values of and can be obtained by using Equations (16-17).” Please see the highlighted in line 122.
It seems that all equation miss punctuation, see for example Eq. 29.
Thank you very much for pointing out this. All punctuation missed in the equations are added.
In the framework of meshless methods especially based on least-squares schemes, I would suggest to mention Moving-Least-Squares WENO schemes. These are indeed particularly useful for both spatial and temporal derivative reconstructions. See for example “Avesani et al, An alternative SPH formulation: ADER-WENO-SPH, Computer Methods in Applied Mechanics and Engineering”, “Antona et al., Towards a High Order Convergent ALE-SPH Scheme with Efficient WENO Spatial Reconstruction, Water, 2021”, “Avesani et al, An alternative smooth particle hydrodynamics formulation to simulate chemotaxis in porous media, Journal of mathematical biology, 2017”.
Thank you very much for the suggestion. These papers are referred in the manuscript as [31-33]. Please also see the highlighted in lines 194-198.
I would add proper labels and legends in the figure. I find difficult sometimes extrapolate all information from the captions of the figures sometimes.
Thank you very much for the suggestion. Labels for Figures 1, 4-6, 8-9 are added.
